# Pro-death NMDA receptor signaling is promoted by the GluN2B C-terminus independently of Dapk1

Jamie McQueen[1,2,3†], Tomás J Ryan[4,5,6,7†], Sean McKay[1,2,3†], Katie Marwick[3], Paul Baxter[1,2,3], Sarah M Carpanini[8,9], Thomas M Wishart[8,9], Thomas H Gillingwater[1,2,9], Jean C Manson[8,9], David J A Wyllie[3], Seth G N Grant[10,11,12], Barry W McColl[1,2,8*], Noboru H Komiyama[10,11,12*], Giles E Hardingham[1,2,3,9*]

[1]UK Dementia Research Institute, University of Edinburgh, Edinburgh, United Kingdom; [2]Edinburgh Medical School, University of Edinburgh, Edinburgh, United Kingdom; [3]Centre for Discovery Brain Sciences, University of Edinburgh, Edinburgh, United Kingdom; [4]School of Biochemistry and Immunology, Trinity College Dublin, Dublin, Ireland; [5]Trinity Biomedical Sciences Institute, Trinity College Dublin, Dublin, Ireland; [6]Trinity College Institute of Neuroscience, Trinity College Dublin, Dublin, Ireland; [7]Florey Institute of Neuroscience and Mental Health, Melbourne Brain Centre, University of Melbourne, Parkville, Australia; [8]The Roslin Institute, University of Edinburgh, Edinburgh, United Kingdom; [9]nPAD MRC Mouse consortium, University of Edinburgh, Edinburgh, United Kingdom; [10]Wellcome Trust Sanger Institute, Hinxton, United Kingdom; [11]Centre for Clinical Brain Sciences, University of Edinburgh, Edinburgh, United Kingdom; [12]Centre for Neuroregeneration, University of Edinburgh, Edinburgh, United Kingdom

*For correspondence: Barry.
McColl@roslin.ed.ac.uk (BWM);
N.Komiyama@ed.ac.uk (NHK);
Giles.Hardingham@ed.ac.uk
(GEH)

†These authors contributed
equally to this work

Competing interests: The
authors declare that no
competing interests exist.

Reviewing editor: Moses V
Chao, New York University
Langone Medical Center, United
States

**Abstract** Aberrant NMDA receptor (NMDAR) activity contributes to several neurological disorders, but direct antagonism is poorly tolerated therapeutically. The GluN2B cytoplasmic C-terminal domain (CTD) represents an alternative therapeutic target since it potentiates excitotoxic signaling. The key GluN2B CTD-centred event in excitotoxicity is proposed to involve its phosphorylation at Ser-1303 by Dapk1, that is blocked by a neuroprotective cell-permeable peptide mimetic of the region. Contrary to this model, we find that excitotoxicity can proceed without increased Ser-1303 phosphorylation, and is unaffected by Dapk1 deficiency in vitro or following ischemia in vivo. Pharmacological analysis of the aforementioned neuroprotective peptide revealed that it acts in a sequence-independent manner as an open-channel NMDAR antagonist at or near the $Mg^{2+}$ site, due to its high net positive charge. Thus, GluN2B-driven excitotoxic signaling can proceed independently of Dapk1 or altered Ser-1303 phosphorylation.

## Introduction

NMDA receptor (NMDAR) -mediated excitotoxicity plays a key role in acute neurological disorders such as stroke and traumatic brain injury, neuronal loss in Huntington's disease, and is also implicated in synapto-toxicity in Alzheimer's disease (*Choi, 1988*; *Lipton and Rosenberg, 1994*; *Berliocchi et al., 2005*; *Lau and Tymianski, 2010*; *Hardingham and Lipton, 2011*; *Parsons and Raymond, 2014*; *Tu et al., 2014*). Most NMDARs are comprised of two obligate GluN1 subunits and two GluN2 subunits (*Furukawa et al., 2005*), with GluN2A and GluN2B predominant in the

forebrain (*Monyer et al., 1994*; *Cull-Candy et al., 2001*; *Traynelis et al., 2010*; *Paoletti, 2011*; *Wyllie et al., 2013*). GluN2 subunits have long, evolutionarily divergent cytoplasmic C-terminal domains (CTDs) which we have shown can differentially associate with signalling molecules (*Martel et al., 2012*; *Ryan et al., 2008, 2013*; *Frank et al., 2016*) and differentially signal to cell death: the CTD of GluN2B (CTD$^{2B}$) potentiates excitotoxicity more strongly than that of GluN2A (*Martel et al., 2012*).

While multiple pathways contribute to excitotoxicity (*Tymianski, 2011*), the mechanism by which CTD$^{2B}$ is thought to potentiate excitotoxicity is upstream of all of them (*Parsons and Raymond, 2014*; *Tu et al., 2010*; *Lai et al., 2014*). The mechanism is centred on Ser-1303 of CTD$^{2B}$, within a region of the CTD unique to GluN2B, and with which CaMKIIα is known to interact and phosphorylate (*Bayer et al., 2001*; *Mao et al., 2014*). It was reported that in response to ischemia or excitotoxic insults, a different kinase, Dapk1, causes Ser-1303 phosphorylation which increases NMDAR-dependent ionic flux (*Tu et al., 2010*). Consistent with this, *Dapk1$^{-/-}$* neurons were reported to be resistant to excitotoxicity, and a cell-permeable peptide mimetic of the CTD$^{2B}$ region around Ser-1303 disrupted Ser-1303 phosphorylation and was neuroprotective (*Tu et al., 2010*).

Given that the GluN2B-Dapk1 pathway is prominent in contemporary models of excitotoxicity (*Parsons and Raymond, 2014*; *Lai et al., 2014*) we sought to investigate this pathway further. Dapk1 has not hitherto emerged from proteomic post-synaptic density screens (*Husi et al., 2000*; *Collins et al., 2006*; *Yoshimura et al., 2004*; *Li et al., 2004*; *Jordan et al., 2004*; *Peng et al., 2004*; *Cheng et al., 2006*; *Bayés et al., 2011*), and we failed to detect it in a recent proteomic analysis of native NMDAR supercomplexes (*Frank et al., 2016*). Moreover, the use of cell-permeable peptides to draw wide-ranging mechanistic conclusions can be problematic without extensive controls. We investigated whether Dapk1-mediated Ser-1303 phosphorylation indeed represents the major reason why CTD$^{2B}$ promotes excitotoxicity signaling better than CTD$^{2A}$, using approaches that include analysis of a new Dapk1 knockout mouse.

## Results

### Excitotoxic insults do not induce GluN2B Ser-1303 phosphorylation

We first examined the influence of excitotoxic conditions on GluN2B Ser-1303 phosphorylation in cortical neurons using a phospho-(Ser-1303)-specific antibody (Millipore 07–398), previously used and validated by several groups (*Jalan-Sakrikar et al., 2012*; *Jia et al., 2013*; *Castillo et al., 2011*).

We confirmed that the antibody is capable of detecting changes in Ser-1303 phosphorylation in neurons: phospho-GluN2B(Ser-1303) levels in cortical neurons, as assayed by western blot using this antibody, are lowered after incubation of cortical neurons with the general kinase inhibitor staurosporine, and increased modestly by a cocktail of phosphatase inhibitors okadaic acid and FK-506 (*Figure 1—figure supplement 1a,b*, *Figure 1—source data 1*). As further evidence of specificity, we found that the antibody completely failed to react with GluN2B in which we had engineered mutations (L1298A/R1300N/S1303D) into the site for a separate study (*Figure 1—figure supplement 1c,d*).

We found that bath application of NMDA at excitotoxic concentrations failed to induce significant Ser-1303 phosphorylation (*Figure 1a and b*, *Figure 1—source data 2*). At the late timepoint (60 min, 50 μM NMDA) we observed a decline in Ser-1303 phosphorylation (*Figure 1a,b*, *Figure 1—figure supplement 1e,f*, *Figure 1—source data 3*) as well as a decline in total levels of GluN2B, consistent with observations of others who have reported partial calpain-mediated cleavage and degradation of the NMDAR CTD (*Dong et al., 2006*; *Gascón et al., 2008*). These observations in DIV10 cortical neurons were also repeated at DIV16 (*Figure 1—figure supplement 1g,h*, *Figure 1—source data 4*). We also saw no increase in Ser-1303 phosphorylation in response to oxygen-glucose deprivation (OGD) (*Figure 1c,d*, *Figure 1—source data 5*), contrary to previous reports (*Tu et al., 2010*).

To determine whether Dapk1 plays any role in the GluN2B Ser-1303 phosphorylation status, we obtained a *Dapk1$^{-/-}$* mouse line, created by the International Mouse Phenotyping Consortium by targeted deletion of exon four on a C57Bl/6 background (the same strain as the Dapk1$^{-/-}$ mouse generated by *Tu et al. (2010)*). The mice had normal fertility, viability and body weight (http://www.mousephenotype.org, MGI:1916885). We confirmed that Dapk1$^{-/-}$ neurons expressed no Dapk1

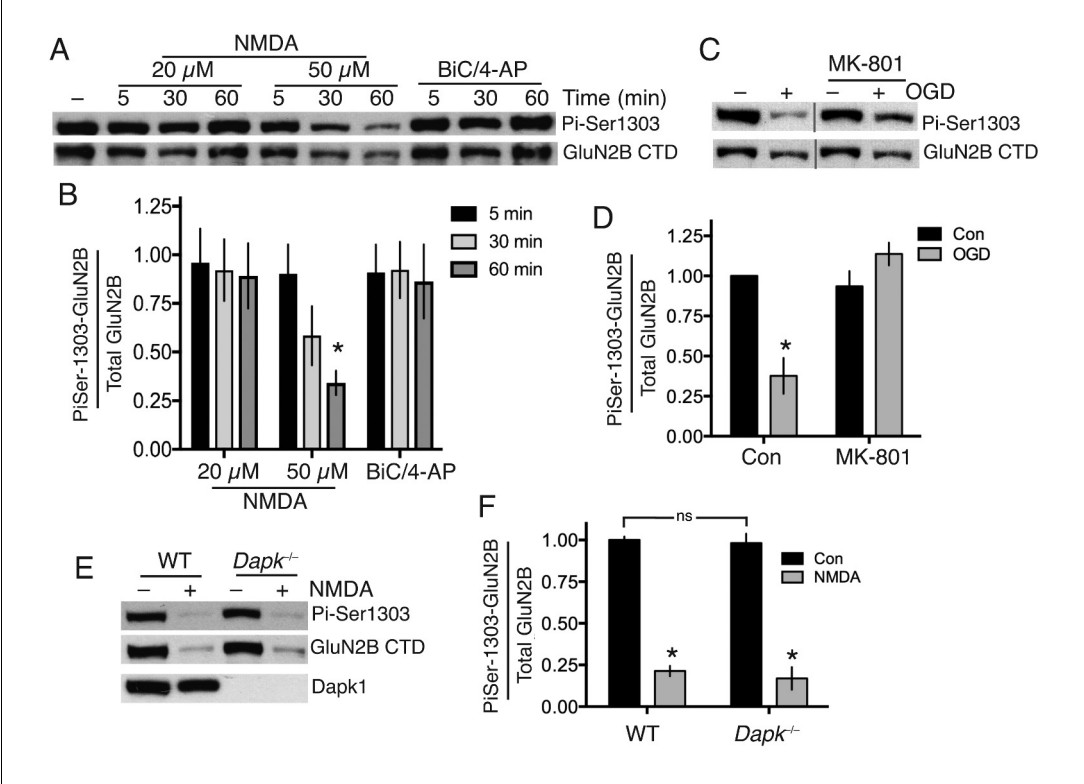

**Figure 1.** Neither Dapk1 nor excitotoxic insults increase GluN2B phosphorylation on Ser-1303. (A,B) Strong excitotoxic insults induce GluN2B Ser-1303 dephosphorylation at later timepoints. Western analysis of extracts from cortical neurons treated as indicated with NMDA or bicuculline (50 µM) plus 4-amino pyridine (250 µM). (F(2,24)=3.904, p=0.034 (Two-way ANOVA). *p=0.0053 (Sidak's post-hoc test; 95% CI of diff: 0.1777 to 1.139, comparison to control without NMDA treatment, N = 3). (C,D) Mimicking ischemic conditions triggers dephosphorylation of GluN2B Ser-1303 in an NMDAR-dependent manner. Oxygen-glucose deprivation (OGD) applied for 120 min ±MK-801 (10 µM). F(1,12)=6.69, p=0.024 (Two-way ANOVA). *p=0.0003 (Sidak's post-hoc test, 95% CI of diff: 0.3289 to 0.9172, N = 4). (E,F) Dapk1 deficiency does not influence basal or NMDA-induced GluN2B Ser-1303 phosphorylation status. Neurons were treated ±50 µM NMDA for 60 min. F(1,10)=345.1, p<0.0001 (Two-way ANOVA, Con vs. NM). *p<0.0001 (both, compared to Con of that genotype, 95% CI of diff: 0.6384 to 0.9342, and 0.6411 to 0.9826 (reading left to right), N = 4 WT, N = 3 KO; with 'N' defined as a distinct culture from a distinct animal). ns: F(1,10)=0.5418, p=0.4786.

The following source data and figure supplement are available for figure 1:

**Source data 1.** Data relating to *Figure 1—figure supplement 1a*.

**Source data 2.** Data relating to *Figure 1b*.

**Source data 3.** Data relating to *Figure 1—figure supplement 1f*.

**Source data 4.** Data relating to *Figure 1—figure supplement 1h*.

**Source data 5.** Data relating to *Figure 1d*.

**Source data 6.** Data relating to *Figure 1f*.

**Figure supplement 1.** (A,B) Neurons were treated with staurosporine (STS, 1 µM) or FK-506 (FK, 5 µM)+okadaic acid (OA, 10 µM) for one hour, after which protein was harvested and western analysis for Phospho- (GluN2B Ser-1303) levels performed.

(*Figure 1e*). We compared GluN2B phospho-Ser-1303 levels in cortical neurons obtained from *Dapk1⁻/⁻* and *Dapk1⁺/⁺* littermates and found no difference in basal levels, nor any difference in the lowered level that we observe at longer periods of NMDA exposure (*Figure 1e,f, Figure 1—source*

*data 6*). Thus in our hands, Dapk1 does not influence GluN2B Ser-1303 phosphorylation status under basal or excitotoxic conditions.

## Excitotoxic and ischemic neuronal death can proceed independently of Dapk1

We next addressed the more general point of the role of Dapk1 in excitotoxic neuronal death. Compared to cortical neurons cultured from their wild-type littermates, we observed no difference in NMDAR-dependent excitotoxic neuronal death in $Dapk1^{-/-}$ neurons at either DIV10 or DIV16 (*Figure 2a*, *Figure 2—source data 1*; *Figure 2b*, *Figure 2—source data 2*) and no difference in OGD-induced neuronal death (*Figure 2c*, *Figure 2—source data 3*), contrary to previous reports. NMDAR currents were also no different in $Dapk1^{-/-}$ vs. $Dapk1^{+/+}$ neurons (*Figure 2d*, *Figure 2— source data 4*).

We then studied the influence of Dapk1 deficiency on ischemic neuronal death in vivo. We employed a model of transient global ischemia model (bilateral common carotid artery occlusion) used previously to show a protective effect of Dapk1 deficiency (*Tu et al., 2010*). Adult mice exposed to a transient (20 min) period of global ischemia showed characteristic selective neuronal death within the hippocampus, particularly the CA1 and CA2 regions. However, $Dapk1^{-/-}$ and $Dapk1^{+/+}$ mice exhibited similar levels of infarction (*Figure 2e–h*, *Figure 2—source data 5*), contrary to previous reports (*Tu et al., 2010*). These observations collectively indicate that excitotoxic and ischemic neuronal death in vitro and in vivo can proceed normally in the absence of Dapk1.

## TAT-NR2B$_{CT}$ is a direct NMDAR antagonist

In support of the Dapk1 hypothesis for CTD$^{2B}$-derived excitotoxicity, a cell-permeable (TAT-fused) peptide mimetic of the GluN2B amino acids 1292–1304 (TAT-KKNRNKLRRQHSY: TAT-NR2B$_{CT}$) was reported to prevent NMDAR-dependent GluN2B Ser-1303 phosphorylation, and excitotoxicity (*Tu et al., 2010*). We observed that 50 µM TAT-NR2B$_{CT}$, the concentration used previously (*Tu et al., 2010*), was toxic to neurons (*Figure 3—figure supplement 1a*, *Figure 3—source data 1*), so we used a concentration 10 times lower (5 µM). We found that 5 µM TAT-NR2B$_{CT}$ completely prevents NMDA-induced excitotoxicity (*Figure 3—figure supplement 1b*, *Figure 3—source data 2*). This was surprising, given that we did not see a role for Dapk1 in excitotoxicity (*Figure 2*). However, further analysis revealed the explanation: at 5 µM, TAT-NR2B$_{CT}$ potently inhibited NMDAR currents, acting immediately and without any need for a preincubation period (*Figure 3a–d*, *Figure 3— source data 3*), and in a manner that was not readily washed out upon removal of peptide (data not shown). TAT-NR2B$_{CT}$ was custom synthesized for our studies by Genscript, and we found that NR2B$_{CT}$(1292–1304)-TAT, a pre-made peptide sold by Merck Millipore was a similarly potent NMDAR antagonist (n = 8, *Figure 3—figure supplement 1c*). A scrambled version of TAT-NR2B$_{CT}$ (TAT-sNR2B$_{CT}$) was neuroprotective and similarly antagonistic at the NMDAR (*Figure 3—figure supplement 1*; *Figure 3—figure supplement 1b*, *Figure 3a–d*). One potential explanation for the NMDAR antagonistic properties of TAT-NR2B$_{CT}$ is the high positive charge of the peptide (+15 at neutral pH). To investigate this, we designed an arginine-rich peptide of high net positive charge (+15 same as TAT-NR2B$_{CT}$) of sequence: RRR TQN RRN RRT SRQ NRR RSR RRR) which strongly antagonized NMDAR currents, and another peptide of net neutral charge (NIN IHD VKV LPG GMI KSN DGP PIL), which had a much weaker effect (*Figure 3a*, *Figure 3—source data 3*). Taken together these data suggest that the net positive charge of TAT-NR2B$_{CT}$ is primarily responsible for its NMDAR-antagonistic properties. We hypothesized that TAT-NR2B$_{CT}$ may be an open channel blocker drawn partly into the pore by its high net positive charge and bind near the internal Mg$^{2+}$ binding site. Consistent with this, the presence of Mg$^{2+}$ (a pore blocker) reduced the effectiveness of TAT-NR2B$_{CT}$'s antagonism (*Figure 3e*, *Figure 3—source data 4*). Another prediction of this hypothesis is that TAT-NR2B$_{CT}$ would be more effective at antagonising NMDARs under open-channel conditions. To test this, NMDAR currents were measured, after which neurons were incubated in TAT-NR2B$_{CT}$ (5 µM) for 60 s in the presence of zero glycine +100 µM AP5 ('block (1)") to ensure minimal channel opening (closed channel conditions). 60 s was chosen because under conditions of NMDAR agonism this is sufficient time to achieve maximal blockade. After 60 s, both TAT-NR2B$_{CT}$ and AP5 were removed from the bathing medium and NMDAR currents were subsequently re-measured (150 µM NMDA +100 µM glycine, zero Mg$^{2+}$). The very slow off-rate of the TAT-NR2B$_{CT}$

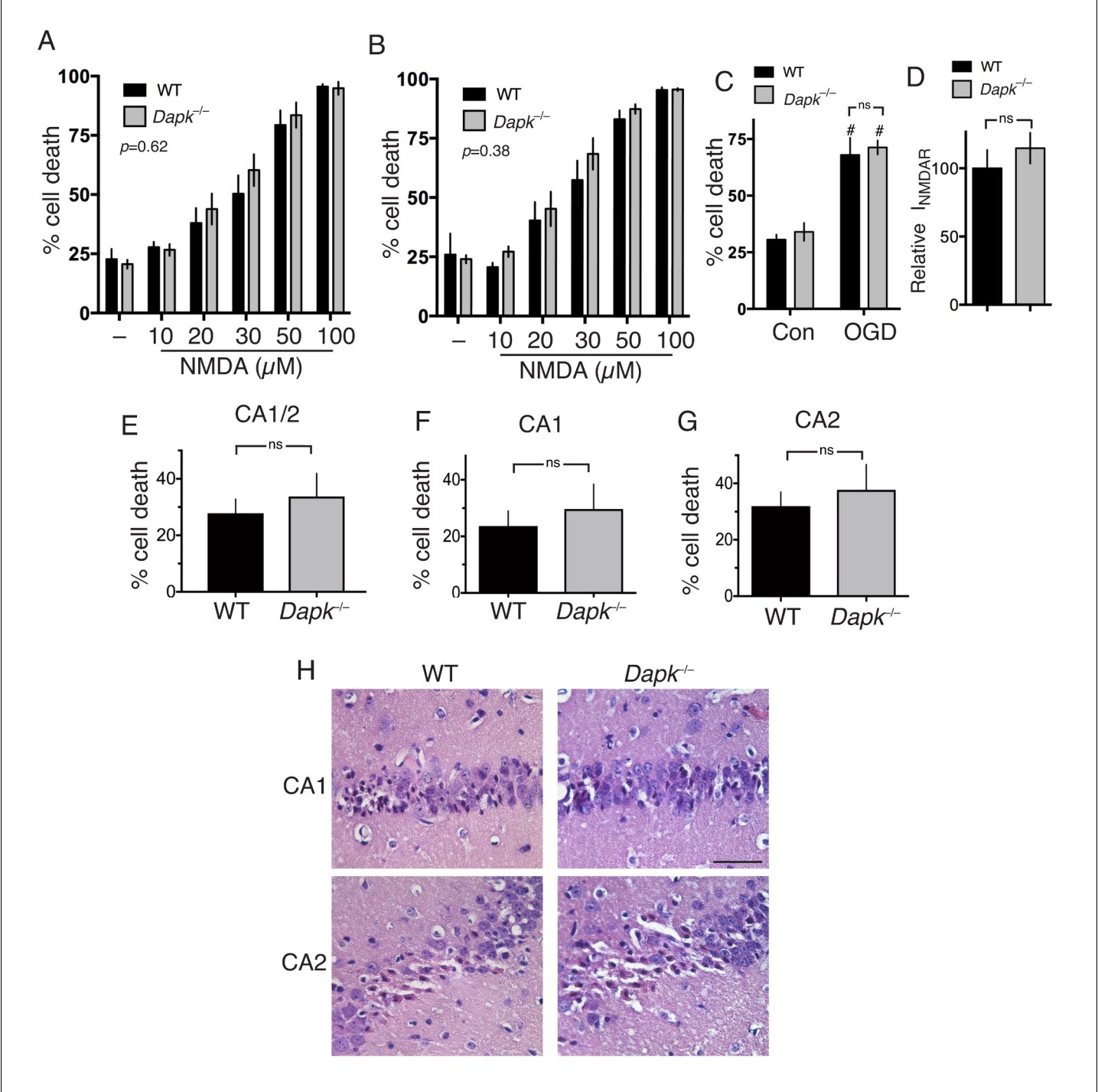

**Figure 2.** Excitotoxic and ischemic insults are not ameliorated by Dapk1 deficiency. (A, B) NMDA-induced neuronal death is independent of Dapk1. Cortical neurons at DIV10 (A) or DIV16 (B) were treated as indicated for 1 hr, with neuronal death assessed at 24 hr. The p values relate to a two-way ANOVA test of differences between WT and Dapk$^{-/-}$ neurons (F(1,10) = 0.2676, n = 6 WT, 6 KO (DIV10); F(1,7)=0.8871, 2-way ANOVA, n = 4 WT, 5 KO (DIV16)). For each condition/genotype combination, 800–1000 cells were analysed per biological replicate. (C) OGD-induced neuronal death is independent of Dapk1. Cortical neurons at DIV10 were subjected to OGD for 120 min, before being returned to control medium. Neuronal death was assessed at 24 hr. No genotype-dependent difference was observed (F (1,12)=0.5062, p=0.490, but a strong influence of OGD was observed: F (1,12) =63.54, p<0.0001, two-way ANOVA. #p=0.0002, 0.0002 (reading left to right); Sidak's post-hoc test comparing control to OGD condition (n = 4 WT, n = 4 KO). (D) Dapk1 deficiency does not influence NMDAR currents. NMDAR currents were measured in n = 16 WT cells (from four separate cultures) and n = 25 KO cells (from six separate cultures). Currents were normalized to the mean current recorded from WT cells recorded on that precise day. p=0.411 (t = 0.831, df = 39), unpaired t-test. (E–G) Dapk1 deficiency does not influence vulnerability to ischemia in vivo. Adult male age-matched mice

*Figure 2 continued on next page*

*Figure 2 continued*

(n = 14 WT; n = 16 KO) were subjected to 20 min bilateral common carotid artery occlusion, sacrificed at 3 d, and pathology analysed. CA1/2 (**E**): p=0.555 (t = 0.598, df = 28); CA1 (**F**) :p=0.572 (t = 0.572, df = 28), CA2 p=0.592(G, t = 0.543, df = 28). Scale bar = 50 μm.

The following source data is available for figure 2:

**Source data 1.** Data relating to *Figure 2a*.
**Source data 2.** Data relating to *Figure 2b*.
**Source data 3.** Data relating to *Figure 2c*.
**Source data 4.** Data relating to *Figure 2d*.
**Source data 5.** Data relating to *Figure 2e–g*.

enabled the peptide's effects on currents to be measured in the absence of the peptide in the medium. TAT-NR2B$_{CT}$ was then applied for a second 60 s ('block (2)') either under the same 'closed channel conditions' or under 'open channel conditions' (100 μM glycine, 150 μM NMDA). After 60 s TAT-NR2B$_{CT}$ was removed from the bathing medium and NMDAR currents measured for a third time. The NMDAR current remaining at the second and third measurements was calculated as a fraction of the initial current. We found that the second peptide incubation (block (2)) significantly increased the proportion of NMDAR inhibition when it occurred under open-channel conditions, but not under closed channel conditions (*Figure 3f*, *Figure 3—source data 5*), further evidence in favour of a pore-centred binding site for TAT-NR2B$_{CT}$. Thus, the unintended NMDAR antagonistic properties TAT-NR2B$_{CT}$ explain its anti-excitotoxic effects.

## Discussion

Dapk1-mediated GluN2B Ser-1303 phosphorylation, and consequent enhancement of toxic Ca$^{2+}$ influx through extrasynaptic NMDARs lies at the heart of current models of excitotoxicity and of the central role of the GluN2B CTD in this process (*Parsons and Raymond, 2014*; *Lai et al., 2014*), but our study suggests that this needs to be re-appraised. Our observations regarding the (lack of) impact of Dapk1 gene deletion on neuronal vulnerability to excitotoxic and ischemic conditions is at odds with previous reports (*Tu et al., 2010*). The *Dapk1$^{-/-}$* mouse that we used was generated independently of the one generated by Tu et al., although there is no *a priori* reason why the two lines should behave differently at this fundamental level, particularly given the very similar genetic background (C57BL/6).

The potent inhibition of NMDAR currents by TAT-NR2B$_{CT}$ at a concentration up to 100 times lower than that used previously (*Tu et al., 2010*) suggests a simple explanation for its neuroprotective effects independent of Dapk1. We are unable to explain why we observed similar effects of TAT-NR2B$_{CT}$ and its scrambled version, while a selective effect of TAT-NR2B$_{CT}$ was previously reported (*Tu et al., 2010*). Both scrambled versions employed had identical sequences, and the potent NMDAR antagonistic properties of our scrambled peptide are consistent with its neuroprotective properties.

The basis for CTD$^{2B}$-mediated excitotoxicity (*Martel et al., 2012*) remains incompletely understood. Exchanging the CTD of GluN2B with that of GluN2A by targeted exon exchange reduces vulnerability to excitotoxicity (*Martel et al., 2012*), without altering the proportion of NMDARs at synaptic vs. extrasynaptic sites, an important factor in excitotoxicity (*Hardingham and Bading, 2010*). Moreover, performing the reciprocal swap increases vulnerability (SM and GEH, unpublished observations), strongly supportive of a key role for CTD$^{2B}$. An ongoing avenue of investigation is focussed on understanding the extent to which the composition of the native NMDAR signaling complex is altered by manipulating the endogenous GluN2 CTDs in our panel of knock-in mice. We hypothesize that alterations to the complex may disturb signaling to pro-death events such as NO production, NADPH oxidase activation, oxidative stress, calpain activation and mitochondrial Ca$^{2+}$

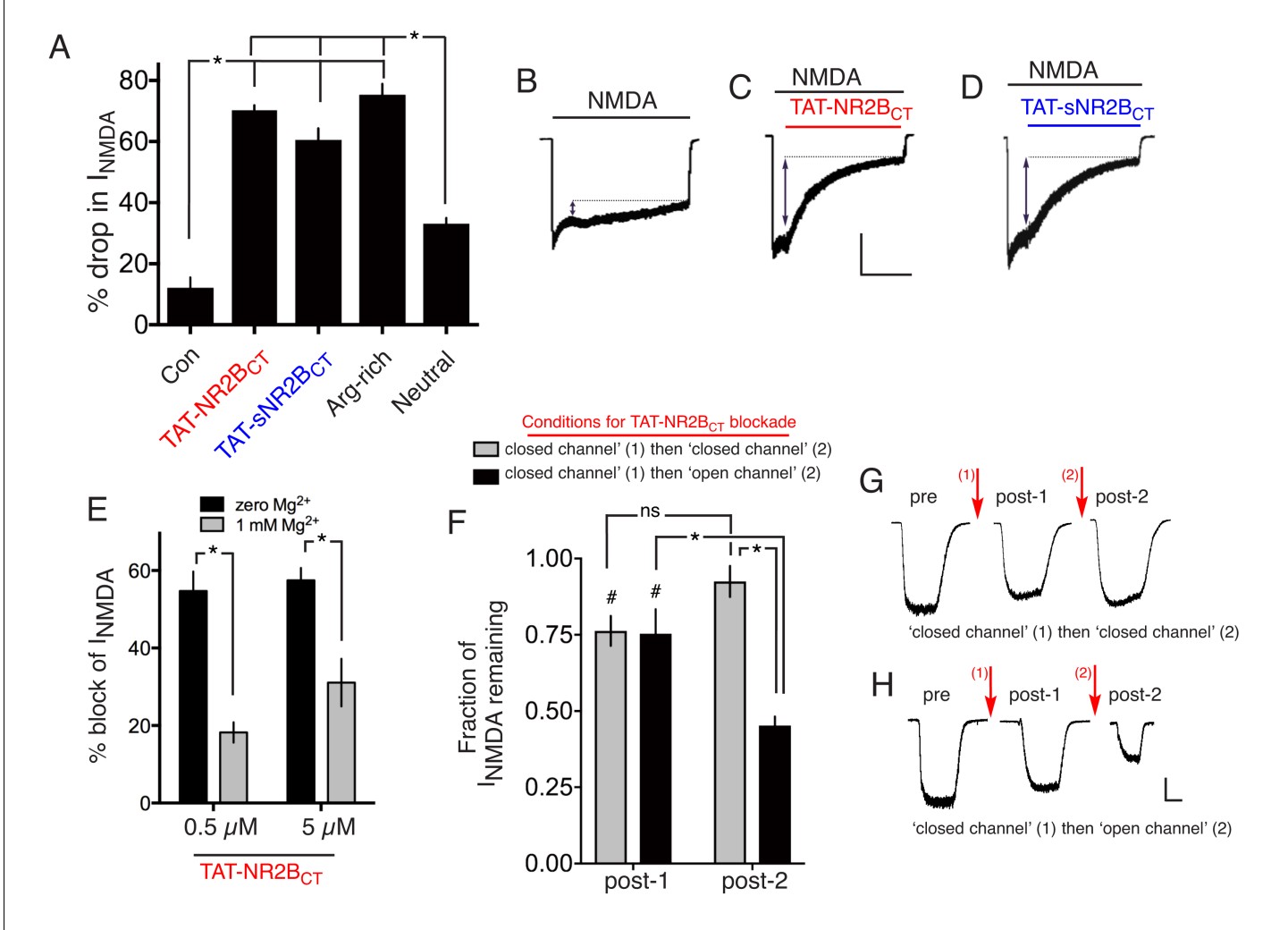

**Figure 3.** Both TAT-NR2B$_{CT}$ and TAT-sNR2Bs$_{CT}$ are direct NMDAR antagonists. (**A–D**) Both TAT-NR2B$_{CT}$ and TAT-sNR2B$_{CT}$ (scrambled version of TAT-NR2B$_{CT}$) immediately antagonize NMDAR currents upon extracellular exposure. NMDA-induced currents were recorded under whole-cell voltage clamp, with the indicated peptides (at 5 µM) applied approximately 5 s after NMDA (to allow NMDAR currents to reach steady state). Arg-rich refers to the arginine-rich positively charged peptide; Neutral refers to the neutral peptide-see main text for sequences of these as well as TAT-NR2B$_{CT}$ and TAT-sNR2B$_{CT}$. NMDA-induced NMDAR currents were monitored for a further 45 s and the percentage drop in currents calculated, compared to no peptide at all (Con) which represents a measure of natural desensitization over this period. $p < 0.0001$ (one-way ANOVA). *$p < 0.0001$, Sidak's post-hoc test ($n = 8$ of all conditions). Example traces shown in (**B**) (**C**) and (**D**). Scale bar: 15 s, 500 pA. (**E**) NMDAR antagonism by TAT-NR2B$_{CT}$ is inhibited by Mg$^{2+}$ blockade. NMDAR currents were measured, after which neurons were incubated in TAT-NR2B$_{CT}$ (5 or 0.5 µM) for 60 s in the presence or absence of 1 mM Mg$^{2+}$, after which NMDAR currents were measured again (in zero Mg$^{2+}$, no peptide). $p < 0.0001$ (one-way ANOVA, $F_{(1,22)}=47.16$ (effect of [Mg$^{2+}$])). *$p < 0.0001$, $p = 0.0007$, Sidak's post-hoc test (zero Mg$^{2+}$: $n = 6$ (0.5 µM), $n = 7$ (5 µM); 1 mM Mg$^{2+}$: $n = 6$ (0.5 µM), $n = 7$ (5 µM)). (**F–H**) NMDAR antagonism by TAT-NR2B$_{CT}$ is more effective on open channels. See main text for experimental details. $p < 0.0001$ (two-way ANOVA, comparing initial current with subsequent measurements: $F_{(2, 45)}=25.22$. $p < 0.0001$ (two-way ANOVA, comparing 'closed-then-closed' protocol (grey bars, $n = 8$) with 'closed-then-open' protocol (black bars, $n = 9$): $F_{(1, 45)}=18.26$. #$p=0.003$, $0.0009$ (Sidak's post-hoc test), comparing to initial currents. *$p < 0.0001$ (Sidak's post-hoc tests), comparisons indicated. (**G**) shows example recordings taken during the consecutive 'closed' then 'closed' channel protocol. (**H**) shows example recordings taken during the consecutive 'closed' then 'closed' channel protocol. Scale bar = 1 s, 250 pA.

The following source data and figure supplement are available for figure 3:

**Source data 1.** Data relating to *Figure 3—figure supplement 1a*.
**Source data 2.** Data relating to *Figure 3—figure supplement 1b*.
**Source data 3.** Data relating to *Figure 3a*.
*Figure 3 continued on next page*

*Figure 3 continued*

**Source data 4.** Data relating to *Figure 3e*.
**Source data 5.** Data relating to *Figure 3f*.
**Figure supplement 1.** (A) Neurons were treated where indicated with 50 μM TAT-NR2B$_{CT}$ for 1 hr, with death assessed after 24 hr.

overload (*Parsons and Raymond, 2014*; *Lai et al., 2014*; *Hardingham and Bading, 2010*; *Bano and Nicotera, 2007*; *Duchen, 2012*; *Nakamura and Lipton, 2011*; *Bell and Hardingham, 2011*; *Panayotis et al., 2015*). Of note, we recently showed that the CTD of GluN2B (as opposed to that of GluN2A) is critically required for formation of 1.5 MDa NMDAR supercomplexes (*Frank et al., 2016*). Thus, regions unique to GluN2B (of which the CaMKII site is one) play a role in higher order signal complex assembly and this may underlie the key role of CTD$^{2B}$ in downstream excitotoxicity (*Martel et al., 2012*).

## Materials and methods

### Neuronal culture, Dapk$^{-/-}$ mice, induction of excitotoxicity and oxygen-glucose deprivation

Cortical mouse neurons were cultured as described (*Bell et al., 2011a*) at a density of between 9–13 $\times$ 10$^4$ neurons per cm$^2$ (*Lipton and Rosenberg, 1994*) from E17.5 mice with Neurobasal growth medium supplemented with B27 (Invitrogen, Paisley, UK). Stimulations of cultured neurons were done in most cases after a culturing period of 9–11 days during which neurons develop a network of processes, express functional NMDA-type and AMPA/kainate-type glutamate receptors, and form synaptic contacts. Other experiments were performed at DIV 16. Dapk$^{-/-}$ mice (colony name: H-Dapk1-B11-TM1B, MGI Allele Name: Dapk1tm1b(EUCOMM)Hmgu, RID:MGI:5756958) were generated by MRC Harwell from targeted ES cells made by The European Conditional Mouse Mutagenesis Program, as part of the International Mouse Phenotyping Program. Dapk$^{-/-}$ genotyping reactions were performed using the following primers: A = 5 AGAGAAACTGAGGCACCTGG −3', B =, 5'-CATCCAAAGTCCACAGCCAC-3', C = 5'-CCAGTTGGTCTGGTGTCA-3' Primer pair A-B recognised the wild-type allele and amplified a product of 322 bp. Primer pair B-C recognised the mutant allele corresponding to a product of 468 bp. PCR reactions were performed using the following cycling conditions: 15 min at 95°C; 36 cycles of 45 s at 94°C, 45 s at 60°C and 1 min at 72°C; and 10 min at 72°C.

To apply an excitotoxic insult, neurons were first placed overnight into a minimal defined medium (*Baxter et al., 2011*) containing 10% MEM (Invitrogen), 90% Salt-Glucose-Glycine (SGG) medium ( [*Bading et al., 1993*]; SGG: 114 mM NaCl, 0.219% NaHCO$_3$, 5.292 mM KCl, 1 mM MgCl$_2$, 2 mM CaCl$_2$, 10 mM HEPES, 1 mM Glycine, 30 mM Glucose, 0.5 mM sodium pyruvate, 0.1% Phenol Red; osmolarity 325 mosm/l, [*Papadia et al., 2005*]). Where used, TAT-NR2B$_{CT}$ or TAT-sNR2B$_{CT}$ (5 μM) was incubated for 1 hr prior to the excitotoxic insult. Neurons were then treated with NMDA (Tocris Bioscience, Bristol, UK) at the indicated concentrations for 1 hr, after which medium was changed to NMDA-free. After a further 23 hr, neurons were fixed and subjected to DAPI staining and cell death quantified by counting (blind) the number of shrunken, pyknotic nuclei as a percentage of the total. To induce oxygen-glucose deprivation, a previously described approach was used (*Bell et al., 2011b*, *2015*). Briefly, cells were washed and incubated in glucose-free SGG (see formulation above, but with glucose replaced by mannitol) that had been previously degassed with 95% N$_2$-5% C0$_2$. The cells were then placed in an anoxic modular incubator chamber for 120 min (as compared to cells washed and incubated in normoxic glucose-containing SGG). For analysis of excitotoxicity, approximately 800–1000 cells were analysed per condition, per replicate (repeated across several replicates), the observer blind to genotype and experimental condition.

## Electrophysiological recording and analysis

Coverslips containing cortical neurons were transferred to a recording chamber perfused (at a flow rate of 3–5 ml/min) with an external recording solution composed of (in mM): 150 NaCl, 2.8 KCl, 10 HEPES, 2 $CaCl_2$, 1 $MgCl_2$, 10 glucose and 0.1 glycine, pH 7.3 (320–330 mOsm). Patch-pipettes were made from thick-walled borosilicate glass (Harvard Apparatus, Kent, UK) and filled with a K-gluconate-based internal solution containing (in mM): potassium gluconate 141, NaCl 2.5, HEPES 10, EGTA 11; pH 7.3 with KOH. Electrode tips were fire-polished for a final resistance ranging between 4–8 M$\Omega$. Currents were recorded at room temperature (21 ± 2°C) using an Axopatch 200B amplifier (Molecular Devices, Union City, CA). Neurons were voltage-clamped at –60 mV, and recordings were rejected if the holding current was greater than –100 pA or if the series resistance drifted by more than 20% of its initial value (<25 M$\Omega$). All NMDA currents were evoked in $Mg^{2+}$-free external recording solution (in which $MgCl_2$ was substituted with 2 mM NaCl) by 150 µM NMDA +100 µM glycine except *Figure 2D* where 50 µM NMDA +100 µM glycine was used. Whole-cell currents were analyzed using WinEDR v3.2 software (John Dempster, University of Strathclyde, UK). The approximate number of cells to be recorded was estimated in order to detect a 25% difference in the parameter under study, powered at 80%, based on the standard deviation of data previously published by the laboratory (*Martel et al., 2012*; *Puddifoot et al., 2012*; *Hardingham et al., 2007*).

To determine the the competing effect of $Mg^{2+}$ and TAT-NR2B$_{CT}$, whole cell NMDA currents were recorded (as described above) followed by the inclusion of TAT-NR2B$_{CT}$ ± $Mg^{2+}$in the recording solution for a blocking period of 60 s. The whole cell NMDA current was then re-assessed, and the percentage block calculated. TAT-NR2B$_{CT}$ was not included when NMDA currents were re-assessed; this may have led to a small washout but we deemed this as negligible due to the slow-off rate of TAT-NR2B$_{CT}$. To investigate the use dependency of TAT-NR2B$_{CT}$, we minimized the possibility of the NMDAR channel opening by spontaneous release of glutamate by removing glycine from the ACSF and co-applying 100 µM AP5. Glycine was added back to the ACSF to measure NMDA currents and to facilitate the block of TAT-NR2B$_{CT}$ in the open channel configuration.

## Western blotting

Western blotting was performed as described (*Baxter et al., 2015*). In order to minimize the chance of post-translational modifications during the harvesting process, neurons were lysed immediately after stimulation in 1.5x LDS sample buffer (NuPage, Life Technologies) and boiled at 100°C for 10 min. Approximately 10 µg of protein was loaded onto a precast gradient gel (4–16%) and subjected to electrophoresis. Western blotting onto a PVDF membrane was then performed using the Xcell Surelock system (Invitrogen) according to the manufacturer's instructions. Following the protein transfer, the PVDF membranes were blocked for 1 hr at room temperature with 5% (w/v) non-fat dried milk in TBS with 0.1% Tween 20. The sample size was calculated based on previous experimental observations of reporting the effect and standard deviation of NMDA-induced Ser-1303 phosphorylation (*Tu et al., 2010*). The membranes were incubated at 4°C overnight with the primary antibodies diluted in blocking solution: Anti phospho-(Ser-1303) GluN2B (1: 2000, Millipore), anti-Dapk1 (1:8000, Sigma), anti-GluN2B (C-terminus, 1:8000, BD Transduction Laboratories), anti-beta actin (1:200000, Abcam). For visualisation of Western blots, HRP-based secondary antibodies were used followed by chemiluminescent detection on Kodak X-Omat film. Western blots were digitally scanned and densitometric analysis was performed using Image J. All analysis of GluN2B phosphorylation was normalized to total GluN2B.

## Bilateral common carotid artery occlusion

Mice were housed in individually-ventilated cages (in groups of up to five mice) under specific pathogen-free conditions and standard 12 hr light/dark cycle with unrestricted access to food and water. All experiments using live animals were conducted under the authority of UK Home Office project and personal licences and adhered to regulations specified in the Animals (Scientific Procedures) Act (1986) and Directive 2010/63/EU and were approved by both The Roslin Institute's and the University of Edinburgh's Animal Welfare and Ethics Committees. Experimental design, analysis and reporting followed the ARRIVE guidelines (https://www.nc3rs.org.uk/arrive-guidelines) where possible. The sample size was calculated based on the experimental observations of reporting the effect and standard deviation of BCCAO-induced neuronal loss in both wild-type and *Dapk1*$^{-/-}$(*Tu et al., 2010*),

whose experimental observations using n = 7 per genotype we retrospectively calculated were powered at >99%.

Transient bilateral common carotid artery occlusion (BCCAO) was performed in *Dapk1$^{-/-}$* and wild-type male control mice under isoflurane anaesthesia (with $O_2$ and $N_2O$). The operator was unaware of genotype. Core body temperature was maintained at 37 ± 0.5℃ throughout the procedure with a feedback controlled heating blanket (Harvard Apparatus, UK). Both common carotid arteries were exposed and dissected from surrounding tissues and occluded by application of an aneurysm clip for 20 min. Clips were removed, the neck wound sutured and topical local anaesthetic (lidocaine/prilocaine) was applied. Mice were recovered on a heated blanket for 4–6 hr and then returned to normal housing. After a 3 day recovery, mice were anaesthetised and perfused transcardially with saline followed by 4% paraformaldehyde. Brains were removed and rostral and caudal blocks prepared using a brain matrix (Harvard Apparatus). Blocks were post-fixed in 4% paraformaldehyde for 24 hr and processed to paraffin blocks. Sections (6 μm) were cut on a microtome (Leica) and stained with haematoxylin and eosin. Ischaemic neuronal death was quantified in the CA1 and CA2 regions of the hippocampus which are the most sensitive regions in this model. Ischaemic (dead) neurons were identified morphologically in two regions of interest (ROIs) in CA1 and the entire CA2 bilaterally. Data are expressed as the number of dead neurons as a % of total neurons in the ROI and show the mean of both hemispheres for each region. All processing and analysis was performed with the operator blind to genotype.

### Statistical analysis, equipment and settings

Statistical testing involved a 2-tailed paired Student's t-test, or a one- or two-way ANOVA followed by an appropriate post-hoc test, as indicated in the legends. Cell death analyses for both in vitro and in vivo experiments were performed blind to the genotype/experimental condition. For all cell death, western blot analyses and in vitro and in vivo cell death experiments, the value of 'N' was taken as the number of independent biological replicates, defined as independently performed experiments on material derived from different animals. For western blots, we used chemiluminescent detection on Kodak X-Omat film, and linear adjustment of brightness/contrast applied (Photoshop) equally across the image, maintaining some background intensity. In any cases where lanes from non-adjacent lanes are spliced together,lanes are always from the same blot, processed in the same way, and the splicing point is clearly marked. Pictures of cells were taken on a Leica AF6000 LX imaging system, with a DFC350 FX digital camera.

## Acknowledgements

We thank Michelle Stewart, Roland Quinney and the team at MRC Harwell, the wider International Mouse Phenotyping Consortium, and the MRC Neurodegenerative Processes of Ageing and Disease (nPAD) mouse network for the generation, supply and import of Dapk$^{-/-}$ mice. We also thank Kathryn Elsegood and David Fricker for mouse colony management and genotyping. This work is funded by the MRC, Alzheimer's Research UK, Alzheimer's Society, the BBSRC (Roslin Institute strategic programme grant - BB/J004332/1), the Wellcome Trust and the European Commission.

## Additional information

### Funding

| Funder | Grant reference number | Author |
|---|---|---|
| Wellcome | WT088156 | Giles E Hardingham |
| Medical Research Council | MRC_G0902044 | Jamie McQueen<br>Sean McKay<br>Paul Baxter<br>Giles E Hardingham |

The funders had no role in study design, data collection and interpretation, or the decision to submit the work for publication.

## Author contributions

JM, KM, Data curation, Formal analysis, Investigation; TJR, Resources, Writing—review and editing; SM, Formal analysis, Investigation, Writing—review and editing; PB, Conceptualization, Supervision, Funding acquisition, Writing—original draft, Writing—review and editing; SMC, Resources, Supervision, Project administration; TMW, Resources, Supervision, Writing—review and editing; THG, Conceptualization, Supervision; JCM, Supervision, Project administration; DJAW, NHK, Supervision, Project administration, Writing—review and editing; SGNG, Resources, Supervision; BWM, Formal analysis, Supervision, Investigation; GEH, Conceptualization, Supervision, Funding acquisition, Writing—original draft, Project administration

## Author ORCIDs

David J A Wyllie, http://orcid.org/0000-0002-4957-6049
Seth G N Grant, http://orcid.org/0000-0001-8732-8735
Barry W McColl, http://orcid.org/0000-0002-0521-9656
Giles E Hardingham, http://orcid.org/0000-0002-7629-5314

## Ethics

Animal experimentation: All experiments using live animals were conducted under the authority of UK Home Office project and personal licences and adhered to regulations specified in the Animals (Scientific Procedures) Act (1986) and Directive 2010/63/EU and were approved by both The Roslin Institute's and the University of Edinburgh's Animal Welfare and Ethics Committees. Experimental design, analysis and reporting followed the ARRIVE guidelines (https://www.nc3rs.org.uk/arrive-guidelines) where possible. Animal experimentation: Animals used in this study were treated in accordance with UK Animal Scientific Procedures Act (1986). The relevant Home Office project licences are P1351480E and 60/4407, and the use of genetically modified organisms approved by local committee reference SBMS 13_007.

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
