## [Decision Letter]

Thank you for submitting your article "Pro-death NMDA receptor signaling is promoted by the GluN2B C-terminus independently of DAPK1" for consideration by *eLife*. Your article has been reviewed by three peer reviewers, and the evaluation has been overseen by a Reviewing Editor and Richard Aldrich as the Senior Editor. The following individuals involved in review of your submission have agreed to reveal their identity: Suzanne Zukin (Reviewer #2).

The reviewers have discussed the reviews with one another and the Reviewing Editor has drafted this decision to help you prepare a revised submission.

Summary:

The study by McQueen et al. contains an important set of data calling into question the dominant notion that phosphorylation of the GluN2B subunit of the NMDA receptor by DAPK1 is a critical aspect of excitotoxic neuronal death of glutamatergic neurons. Using a combination of biochemistry, histology, Ca^2+^ imaging, and whole-cell recordings, McQueen, et al. overturn a key hypothesis as to how NMDA receptors (NMDARs) cause neuronal death. The authors show that, contrary to a paper published by Tu et al. (Cell, 2010), neither activation of death-associated protein kinase 1 (DAPK1) nor excitotoxic insults promote phosphorylation of GluN2B at Ser1303, a known CaMKII binding site and identified site of DAPK1 phosphorylation under ischemic and excitotoxic conditions. They further show that genetic deletion of DAPK1 does not substantially alter ischemia nor OGD-induced neuronal death.

These are important findings since they put a serious hole in the prevailing hypothesis for the nature of NMDAR-mediated neurotoxicity. The short report is therefore an appropriate format for the presentation of these results which are solid and well presented.

Essential revisions:

The full manuscript as submitted doesn't conform to the spirit of the *eLife* "short report". Several experiments have been represented in each of the 3 main figures, and several more in each of the supplemental figures. At times, too little information is given to permit a full understanding of how each experiment was performed.

The most important three pieces of new information are 1) Excitotoxic treatments of cultured neurons do not induce phosphorylation of S1303 on the GluN2B receptor; 2) an independently created DAPK1 knockout mouse mutant shows no difference from wild type in vulnerability to excitotoxic insult, in contrast to previous reports; nor do neurons cultured from the mutant mouse; 3) the tat-NR2B_CT_ peptide, which has been marketed as a potential anti-excitotoxic agent blocks NMDA receptor currents at 5 µM, as does the scrambled control version of the peptide. This finding is particularly important because it accounts for the observed protection from excitotoxicity by this peptide; however, it indicates that the peptide would not be a viable treatment after stroke in humans because direct blockade of the NMDA receptor has dangerous side-effects.

The authors include additional data that lengthens the report and is not necessary for the most important conclusions. For example, the data regarding the GluN2B^ΔCaMKII^ mouse knock-in mutant is incompletely documented and not necessary for the conclusions. A more thorough study would be necessary to document the potential effects of the three mutations, and lack of their effect on NMDAR currents. Data regarding this mutation (Figure 2—figure supplement 1) should be deleted from the paper.

Figure 3, and G regarding calcium currents through NMDARs in the presence of tat-NR2B_CT_ are not necessary for the conclusion and are not properly controlled. There is no example of control calcium currents for 200 sec without addition of the peptides.

If the authors wish to include all of this data, they should write it up as a research paper, with more figures and with each figure fully documented.

Materials and methods should be included as part of the paper, not as supplementary information.

It is important that papers of this nature, which directly contradict previous findings, are properly controlled and powered and there are some issues to address here in this regard:

1) In Figure 1 there is no positive control to show that the phosphorylation selective antibody can detect increases in phosphorylation at Ser 1303 as predicted for the involvement of DAPK1. Such a control is important because ischemic and basal conditions, and therefore signaling pathways, vary considerably between laboratories for both in vitro and in vivo assays.

2) The in vivo ischemia experiments shown in Figure 2 are not sufficiently powered. The variability in cell death between animals is high as shown by the error bars so n=7 seems too low to produce a significant result for any reasonable effect size. More experiments should be added.

3) The control for perfusion of peptide in Figure 3 should include perfusion of an alternative peptide (not scrambled). Furthermore, it would be reasonable to add to the Discussion that additional random peptide sequences are often required as controls in addition to scrambled peptides in protein interaction blocking experiments.

4) In Figure 3 it would be useful to make the comparison of block at +40mV to control since this will likely show no effect of peptide at +40mV.

5) The means of applying oxygen-glucose deprivation (Figure 1) is not described and should be added to the Materials and methods section. Without these methods it is hard to judge whether these experiments are equivalent to those previously performed.

*Full Comments of Reviewer #2 (Suzanne Zukin):*

Using a combination of biochemistry, histology, Ca^2+^ imaging, and whole-cell recordings, McQueen, et al. overturn a key hypothesis as to how NMDA receptors (NMDARs) cause neuronal death. The authors show that, contrary to a paper published by Tu et al. (Cell, 2010), neither activation of death-associated protein kinase 1 (DAPK1) nor excitotoxic insults promote phosphorylation of GluN2B at Ser1303, a known CaMKII binding site and identified site of DAPK1 phosphorylation under ischemic and excitotoxic conditions. They further show that genetic deletion of DAPK1 does not substantially alter ischemia nor OGD-induced neuronal death. This is significant in that Tu et al. reported that DAPK1 serves as a central mediator of stroke damage and that loss of DAKP1 is neuroprotective. Finally, they show evidence that TAT-NR2B_CT_, a peptide reported by the same authors to inhibit DAPK1-dependent phosphorylation of GluN2B at Ser1303, and TAT-sNR2B (scrambled version of TAT-NR2B_CT_) are potent NMDAR antagonists. Collectively, the main findings of the present study overturn the findings of a highly influential paper in the stroke literature. The manuscript is well-written and the experiments appear to be carefully executed. Although positive findings would have strengthened the manuscript, the strong negative findings are likely to be of broad interest.

---

## [Author Response]

*Essential revisions:*

*[…] The authors include additional data that lengthens the report and is not necessary for the most important conclusions. For example, the data regarding the GluN2B^ΔCaMKII^ mouse knock-in mutant is incompletely documented and not necessary for the conclusions. A more thorough study would be necessary to document the potential effects of the three mutations, and lack of their effect on NMDAR currents. Data regarding this mutation (Figure 2—figure supplement 1) should be deleted from the paper.*

We agree that these data are not essential for the major conclusions, and have removed the data.

*Figure 3, and G regarding calcium currents through NMDARs in the presence of tat-NR2B_CT_ are not necessary for the conclusion and are not properly controlled. There is no example of control calcium currents for 200 sec without addition of the peptides.*

We have removed the data from original Figure 3. We considered the (very straightforward) experiment of adding a ‘no peptide’ control. However, given that direct electrophysiological measurements are clear, and that NMDA-induced increased in [Ca^2+^] can theoretically be both due to direct NMDAR-mediated ionic flux as well as indirect due to excitotoxicity-associated Ca^2+^ deregulation, we agree with the editors that the Ca^2+^ imaging data are not necessary.

*If the authors wish to include all of this data, they should write it up as a research paper, with more figures and with each figure fully documented.*

With the data above removed and despite explanations and detail expanded on, the manuscript now conforms better to a Short Report.

*Materials and methods should be included as part of the paper, not as supplementary information.*

This has been corrected.

*It is important that papers of this nature, which directly contradict previous findings, are properly controlled and powered and there are some issues to address here in this regard:*

*1) In Figure 1 there is no positive control to show that the phosphorylation selective antibody can detect increases in phosphorylation at Ser 1303 as predicted for the involvement of DAPK1. Such a control is important because ischemic and basal conditions, and therefore signaling pathways, vary considerably between laboratories for both in vitro and in vivo assays.*

The phospho-GluN2B(Ser1303) antibody is likely to be targeting the correct region, since immunoreactivity is lost in the GluN2B^∆CaMKII^ mouse with mutations at 1298, 1300 and 1303 (Figure 1—figure supplement 1). Moreover, phospho-GluN2B(Ser-1303) levels in cortical neurons, as assayed by western blot using this antibody, are lowered after incubation of cortical neurons with the general kinase inhibitor staurosporine, and increased by a cocktail of phosphatase inhibitors okadaic acid and FK-506. This suggests that the antibody is capable of detecting changes in Ser-1303 phosphorylation, and the fact that these are not altered in DAPK^–/–^ neurons under any conditions suggest that DAPK1 is not the Ser-1303 kinase. These data are now shown in Figure 1—figure supplement 1.

*2) The in vivo ischemia experiments shown in Figure 2 are not sufficiently powered. The variability in cell death between animals is high as shown by the error bars so n=7 seems too low to produce a significant result for any reasonable effect size. More experiments should be added.*

We have performed further experiments, doubling the animals involved, and still see no protection or trend towards protection in either CA1 or CA2 regions, or averaging across the regions.

*3) The control for perfusion of peptide in Figure 3 should include perfusion of an alternative peptide (not scrambled). Furthermore, it would be reasonable to add to the Discussion that additional random peptide sequences are often required as controls in addition to scrambled peptides in protein interaction blocking experiments.*

We feel that the use of a no-peptide control is useful in determining the impact of TAT-NR2B_CT_ on currents at the concertation used by Tu et al. (5 µM), and that the similarly large effect of scrambled TAT-sNR2B_CT_ shows that the effect on currents is not sequence-specific, contrary to Tu et al.’s hypothesis. However, we agree that using alternative peptides (also at 5 µM) may provide clues as to whether *any* peptide can inhibit NMDAR currents at those concentrations, or whether the properties of the amino acids play a role. We have included data for two further peptides-both at 5 µM and both 24 amino acids long (as with TAT-NR2B_CT_) designed to determine whether the high positive charge of TAT-NR2B_CT_ and TAT-sNR2B_CT_ could contribute to its NMDAR-antagonistic properties. We designed an arginine-rich peptide of high net positive charge (+15-same as TAT-NR2B_CT_) of sequence: RRR TQN RRN RRT SRQ NRR RSR RRR) which strongly antagonized NMDAR currents (Figure 3), and another peptide of net neutral charge (NIN IHD VKV LPG GMI KSN DGP PIL), which had a much weaker effect (Figure 3). Taken together these data suggest that the net positive charge of TAT-NR2B_CT_ is primarily responsible for its NMDAR-antagonistic properties.

*4) In Figure 3 it would be useful to make the comparison of block at +40mV to control since this will likely show no effect of peptide at +40mV.*

We performed this experiment as requested and surprisingly found that NMDAR currents in the control actually drifted upwards at +40mV. Since this makes the effects of the peptide hard to interpret we have removed this sub-figure. This data was not essential for our conclusion, since we already show strong evidence that TAT-NR2B_CT_ is blocking the NMDAR in a sequence-insensitive manner due to its high net positive charge (Figure 3) and that it is acting primarily as an open channel blocker (Figure 3) at a site that is in competition with the NMDAR’s Mg^2+^ binding site (Figure 3).

*5) The means of applying oxygen-glucose deprivation (Figure 1) is not described and should be added to the Materials and methods section. Without these methods it is hard to judge whether these experiments are equivalent to those previously performed.*

We apologies for this omission and now include this information in the Materials and methods section. As is standard for this type of experiment, (and as employed by Tu et al) cells were placed in a glucose-free bicarbonate buffer, previously de-oxygenated (with 95% N_2_-5% C0_2_). Our cells were then placed in an anoxic modular incubator chamber before being returned to normoxic, glucose-containing conditions.